# Evaluating Latent Space Robustness and Uncertainty of EEG-ML Models under Realistic Distribution Shifts

**Neeraj Wagh**[1*]**, Jionghao Wei**[1]**, Samarth Rawal**[1]**, Brent Berry**[2]**, Yogatheesan Varatharajah**[1,2*]

[1]University of Illinois at Urbana-Champaign    [2]Mayo Clinic

## Abstract

The recent availability of large datasets in bio-medicine has inspired the development of representation learning methods for multiple healthcare applications. Despite advances in predictive performance, the clinical utility of such methods is limited when exposed to real-world data. This study develops model diagnostic measures to detect potential pitfalls before deployment without assuming access to external data. Specifically, we focus on modeling realistic data shifts in electrophysiological signals (EEGs) via data transforms and extend the conventional task-based evaluations with analyses of a) the model's latent space and b) predictive uncertainty under these transforms. We conduct experiments on multiple EEG feature encoders and two clinically relevant downstream tasks using publicly available large-scale clinical EEGs. Within this experimental setting, our results suggest that measures of latent space integrity and model uncertainty under the proposed data shifts may help anticipate performance degradation during deployment.

## 1 Introduction

The availability of large datasets in bio-medicine has inspired the development of deep representation learning methods for diagnostic applications in multiple health domains, such as cardiology, neurology, and radiology. However, there are increasing examples of failures of such systems during real-world deployment, partly due to dataset or distribution shifts in the deployment setting (1; 2; 3; 4; 5; 6). Currently, the most common approach undertaken to evaluate healthcare machine learning (ML) models is analyzing task performance in a held-out test dataset. The recent advances in covariate-shift detection and adaptation approaches (7; 8; 9; 10; 11) have not been widely adopted in the healthcare ML community because they do not reflect the unique health-data-related challenges. As such, the practical value of many healthcare ML models remains unclear at present without additional assessments of model resilience in deployment settings (12). This gap reflects a major hurdle in translating healthcare ML models that are required to perform consistently under varying noise characteristics, diverse acquisition protocols, and multiple sites and populations. Hence, there is a critical need to evaluate the robustness of healthcare ML models to such shifts prior to deployment.

In this paper, we focus on a particular area of healthcare ML pertaining to learning representations of physiological signals such as electroencephalograms (EEGs). EEGs are commonly used to diagnose various neurological, psychiatric, and sleep disorders, as well as in applications involving brain-machine interfaces. Deep representation learning of raw EEG signals has recently gained popularity because of the availability of large-scale EEG datasets (13) and has shown promise in improving the labor-intensive and error-prone manual process undertaken in clinical EEG reviews (14). Various approaches have been proposed, including fully-supervised deep feature encoders (15; 16; 17), self-supervised encoders (18; 19; 20), and traditional feature-based encoders using power spectral features (21). However, many of those approaches struggle to generalize to previously unseen real-world data, i.e., data acquired from different populations, institutions, or devices (22; 23).

---

*Corresponding authors: {nwagh2, varatha2}@illinois.edu

36th Conference on Neural Information Processing Systems (NeurIPS 2022).

We believe that a multi-pronged approach is necessary to holistically evaluate the robustness of healthcare ML models prior to deployment (Figure 1). First, models of realistic data shifts are needed to assess robustness during model development itself. For instance, realistic EEG data shifts may include differences in behavioral states during EEG acquisition (awake or asleep), the number of EEG sensors used, the EEG reference used, noise during acquisition, analog-to-digital conversion, hardware filter settings, and impedance resulting from electrodes. Second, we argue that robustness evaluations may also benefit from analyzing the properties of the latent embedding space (7). Specifically, quantifying the topological changes in the latent space under perturbations can 1) provide early clues into test set performance when ground truth labels are not available; and 2) indicate failure modes of model interpretation techniques that rely on analyzing similarity properties in the latent space. Third, we highlight the need for estimating the uncertainty in model predictions at the output layer (24). Uncertainty modeling is critical in ensuring fail-safety in healthcare ML as accurate communication of uncertainty allows human experts to ignore the model selectively.

In this study, we evaluated the robustness of several previously proposed EEG feature encoders using the above multi-pronged approach. Specifically, we introduced four EEG signal transformations that model the effects of instrumentation-related real-world EEG variability, namely: hardware band-pass filters, quantization precision, narrow-band impedance noise, and broad-band Gaussian noise. Next, we developed a topological measure of latent space integrity based on Delaunay neighborhood graphs (adapted from (25)) and a Monte Carlo dropout-based approach to estimate predictive uncertainty. In this setting, we then evaluated fully-supervised, self-supervised, and traditional feature-based EEG encoders in both classification and regression tasks. Our major contributions are the following:

1. We developed four domain-guided EEG data shifts (EEG-DS) reflecting instrumentation-related variability observable in real-world deployment.

2. Using EEG-DS, we devised a multi-pronged approach to evaluate the robustness of multiple ML models, including assessments of latent space integrity and predictive uncertainty.

3. Using two large-scale public EEG datasets (TUH (13) and NMT (26)), we show that the proposed approach can help anticipate in-the-wild performance during model development and that popular EEG encoders exhibit variable robustness profiles.

## 2 Related Work

**Generalization to unseen data**: The importance of generalization in healthcare ML is widely acknowledged (1; 5; 6; 27). Prior work in general ML has proposed, a) methods to detect covariate-shift or out-of-distribution samples (7; 28; 29); and b) novel learning approaches that generalize well or adapt to new domains (30; 31; 32; 8; 9; 10). However, the majority of the adaptive approaches require examples from target distributions and therefore have limited utility. Some methods that detect covariate-shift focus on specific properties, such as non-stationarity (28; 29), are simplistic and do not reflect the complexity of healthcare data and shifts. In this paper, we focus on detecting failure modes prior to deployment using topological analysis of the latent space and evaluation of model uncertainty under realistic data shifts. In this context, the closest to our work is (7), where authors used measures such as Wasserstein's distance for detecting data shifts in the latent space for vision and language tasks. Here we develop a more comprehensive topological evaluation of the latent space using Delaunay neighborhood graphs and use it to analyze the robustness of scalp EEG representations. In addition, some recent studies have highlighted the benefits of uncertainty quantification in out-of-distribution settings, in knowing when not to trust a model's predictions (33). Some studies have also indicated that certain types of representation learning approaches exhibit better robustness in terms of predictive uncertainty than others (34). Building on these insights, we utilized a Monte Carlo dropout (MCD)-based uncertainty quantification to analyze the predictive uncertainty of different EEG feature encoders under specific data shifts.

**Data shifts in EEG data:** Several previous studies have analyzed the performance of EEG-ML models using datasets from multiple institutions (22; 23; 35). Apart from reporting differences in model performance, these studies did not investigate approaches to identify failure modes prior to deployment, model data shifts, or generate insights for future studies. Some studies have developed data augmentations for EEG signals to improve the diversity of training data and have demonstrated some success in achieving better task performance (19; 36; 37). While augmentations such as amplitude scaling, time shift, DC shift, and band-stop filtering are useful to increase training data

diversity (19), they do not meet the complexity of real-world EEG signal variability observed in deployment settings. Specifically, real-world EEG variability may arise due to diverse physiological states during acquisition, recording conditions and protocols, and internals of the EEG hardware used. Hence, in this paper, we focus on developing domain-guided data shifts observable in deployment settings as opposed to simpler training time data augmentations.

**EEG feature encoders:** A variety of EEG feature encoders have been proposed including, fully-supervised deep encoders (15; 16; 17), self-supervised encoders (18; 19; 20), and traditional domain-derived feature encoders using power spectral features (21; 38). While the generalization capabilities of the majority of these encoders are unclear, an encoder derived based on self-supervised learning has shown promising trends in generalization when specific domain-derived self-supervision tasks were utilized (18). However, a comprehensive robustness evaluation of the different types of EEG feature encoders under realistic EEG data shifts is currently lacking.

## 3 Background & Methods

In this section, we describe the formulations of multi-channel EEG data, feature encoders, and the proposed multi-pronged robustness evaluation approach including realistic EEG data shifts (EEG-DS), latent space analysis, and uncertainty quantification. The proposed approach (Figure 1) is applicable during model development as it does not require additional data or labels from deployment settings.

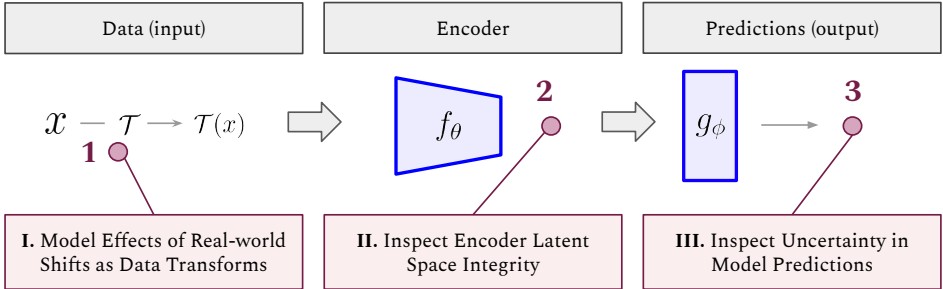

Figure 1: **Multi-pronged robustness evaluation.** Our approach consists of a) implementation of realistic EEG data shifts (Section 3.2), b) latent space analysis (Section 3.3), and c) uncertainty quantification (Section 3.4).

### 3.1 Learning EEG data representations

We use $x_i \in \mathbb{R}^{M \times N}$ to denote the EEG data of a single epoch recorded using $M$ sensors with $N$ samples (i.e., $N =$ time duration $\times$ sampling frequency). Typically, a full EEG recording consists of multiple such non-overlapping epochs. Consider a feature encoder $f_\theta : \mathbb{R}^{M \times N} \to \mathbb{R}^d$ that maps a multi-channel EEG epoch $x_i$ into a $d$-dimensional latent representation $z_i \in \mathbb{R}^d$, where $z_i = f_\theta(x_i)$. In traditional supervised learning, $f_\theta$ is trained jointly with a classification or regression head $g_\phi(.)$ to predict the desired target $y_i$ with the following learning objective, where $\mathcal{L}$ is a loss function.

$$\phi^*, \theta^* = \underset{\phi, \theta}{\operatorname{argmin}} \sum_i \mathcal{L}(y_i, g_\phi(f_\theta(x_i)))$$

However, in representation learning (e.g., self-supervised learning), $f_\theta$ is first pretrained using a different objective to obtain a feature extractor $f_{\theta^*}$, which produces a latent embedding for a given input. Classification or regression heads $g_\phi(.)$ can then be trained using the latent embeddings keeping $f_{\theta^*}$ fixed, where the learning objective takes the following form.

$$\phi^* = \underset{\phi}{\operatorname{argmin}} \sum_i \mathcal{L}(y_i, g_\phi(f_{\theta^*}(x_i)))$$

### 3.2 Realistic data shifts in EEG data

In this study, we focus on modeling data shifts in EEG datasets caused by real-world variability. Consider the training (i.e., source) data domain $p_s(x)$ and the real-world data domain (i.e., target) $p_t(x)$, where $x$ is input data. Covariate shifts are defined when the concept mapping $g_\phi \circ f_\theta : x \to y$

remains unchanged, resulting in the same conditional distribution of labels i.e., $p_s(y|x) = p_t(y|x)$. However, the marginal distributions of data are assumed to change i.e., $p_s(x) \neq p_t(x)$. In this study, we assume no access to external labels or additional data from the real-world. Instead, we synthetically construct the target domain $p_t(x)$ based on an expert understanding of the EEG signal acquisition process via a set of EEG data transformations $\mathcal{T}$. Therefore, within the scope of the current study, $p_t(x) = p_s(\mathcal{T}(x))$, and the proposed robustness assessment approach can be viewed as quantifying the differences between $p_s(x)$ and $p_s(\mathcal{T}(x))$ through a set of metrics $\mathcal{M}$.

The effects of real-world variability on EEG segments can be modeled using a data transformation function $t : \mathbb{R}^{M \times N} \to \mathbb{R}^{M \times N}$, where $\tilde{x} = t(x)$ represents a shifted EEG sample. Real-world EEG data shifts may occur due to differences in populations, behavioral states (e.g., awake, asleep), the number of EEG sensors used, the EEG reference used, noise during acquisition, analog-to-digital conversion, hardware filter settings, and impedance resulting from electrodes. Here, we focus on data shifts induced by acquisition conditions and assume that other variables are controlled. In that context, we develop four EEG transformations: 1) band-pass filter $t_{\text{BP}}$, 2) quantization precision $t_{\text{QP}}$, 3) narrowband impedance noise $t_{\text{IN}}$, and 4) broadband noise $t_{\text{BN}}$ (summarized in Table 1).

Table 1: Realistic EEG data shifts represented as transformations.

| EEG Shift | Definition | Parameters | Description |
|---|---|---|---|
| Band-pass filter | $t_{\text{BP}} := \Psi(x, f_L, f_H)$ | $f_L \in \{0.5, 1\}$ 
 $f_H \in \{25, 30\}$ | $\Psi$: band-pass filter function 
 $f_L$, $f_H$: low and high cut-off filter frequencies |
| Quantization precision | $t_{\text{QP}} := k(x, D)$ | $k : \mathbb{R} \to \mathbb{R}$ 
 $D \in \{6, 8, 12\}$ | $k$: restricts $x$ to D decimal digits |
| Impedance noise | $t_{\text{IN}} := x + \epsilon$ | $\sigma \in \{0.001, 0.01, 0.1\}$ 
 $z_\sigma \sim \mathcal{N}(0, \sigma)$ 
 $\epsilon = \Psi(z_\sigma, 0, 1)$ | $\sigma$: noise strength 
 $z_\sigma$: broadband noise 
 $\epsilon$: 0–1 Hz noise |
| Broadband noise | $t_{\text{BN}} := x + \epsilon$ | $\sigma \in \{0.001, 0.01, 0.1\}$ 
 $\epsilon \sim \mathcal{N}(0, \sigma)$ | $\sigma$: noise strength 
 $\epsilon$: broadband noise |

**Band-pass filter:** EEG device manufacturers commonly implement hardware-level band-pass filters in order to restrict the spectral content of the acquired signal. However, those filter settings may vary between different manufacturers and devices. Hence, we define a data shift $t_{\text{BP}}$ to reflect the effects of different band-pass filter settings such as, 0.5–30 Hz, 1–30 Hz, and 1–25 Hz.

**Quantization precision:** In commercial EEG devices, resource constraints limit the precision of the recorded EEG data and the analog signal is digitized with a variable number of bits. We implement this effect as a truncation of decimal digits to either 6, 8, or 12 points and denote it using $t_{\text{QP}}$.

**Signal noise:** The acquisition of EEG signals is highly susceptible to multiple sources of noise and artifacts: subject movement-related sources in the form of eye blinks, head movement, jaw clenching, etc., environmental sources such as AC power line interference, and instrumentation-related sources such as electrical impedance due to poor sensor contact with the scalp. We use $t_{\text{IN}}$ to model electrical impedance noise due to poor sensor contact or choice of dry/wet electrodes that manifests as narrow-band low-frequency (0–1 Hz) random noise in the EEG signal (39) with different strengths ($\sigma \in \{0.001, 0.01, 0.1\}$). Next, we use $t_{\text{BN}}$ to capture random signal noise that may manifest as broadband (0–45 Hz) noise with different strengths ($\sigma \in \{0.001, 0.01, 0.1\}$).

### 3.3 Topological analysis of the latent space

Here we aim to quantify the distortion in latent space caused by input data transforms, i.e., differences between $z = f_\theta(x)$ and $\tilde{z} = f_\theta(\mathcal{T}(x))$. Some previous studies have utilized distance metrics such as Wasserstein's distance to quantify the differences (7). Here we employ a more comprehensive approach adapted from (40) and (25), where we estimate a manifold that contains point clouds comprising the latent representations $\{z_i\}$ and $\{\tilde{z}_i\}$ and conduct a geometric/topological analysis on this estimated manifold. We estimate the manifold based on Delaunay neighborhood graphs, which connect spatial neighbors in a way that captures local point density as well as global outliers (25).

Suppose we use $Z \in \mathbb{R}^d$ and $\tilde{Z} \in \mathbb{R}^d$ to denote the sets including all embeddings of raw and transformed input data, respectively, and define a set of vertices $\mathcal{V} = \{Z \cup \tilde{Z}\}$. Then, the Voronoi cell of $v \in \mathcal{V}$ is the set of points in $\mathbb{R}^d$ for which $v$ is the closest among $\mathcal{V}$, i.e., $\text{Cell}(v) = \{x \in \mathbb{R}^d \mid \|x - v\| \leq \|x - v_i\| \, \forall v_i \in \mathcal{V}\}$. Then, we define edges $\mathcal{E}$ by connecting points $v_i$ and $v_j$ whose Voronoi cells intersect, i.e., $\mathcal{E} = \{(v_i, v_j) \in \mathcal{V} \times \mathcal{V} \mid \text{Cell}(v_i) \cap \text{Cell}(v_j) \neq \emptyset, v_i \neq v_j\}$. This set of vertices $\mathcal{V}$ and edges $\mathcal{E}$ form the neighborhoods in the Delaunay graph $\mathcal{G} = (\mathcal{V}, \mathcal{E})$.

Intuitively, the points in $\mathcal{V}$ that share a Voronoi cell boundary are considered natural spatial neighbors irrespective of the distance between them. This property of Delaunay neighborhoods enables robust manifold estimation even when point density varies locally and outlier points exist. Notably, these spatial neighbors or edges can emerge in a way that connects points in $Z$ to points in $\tilde{Z}$ (denoted by $R$ and $E$ respectively in (25)). Such edges can be considered 'heterogeneous', as opposed to 'homogeneous' edges which connect two points belonging to the same set. The degree of edge heterogeneity of $\mathcal{G}$ can be measured by a simple *quality* measure (40) $q(\mathcal{G})$ defined in Equation 1.

In Equation 1, $\left|\mathcal{G}^Z\right|_{\mathcal{E}}$ and $\left|\mathcal{G}^{\tilde{Z}}\right|_{\mathcal{E}}$ denote the number of homogeneous edges within $\mathcal{G}$ belonging to $Z$ and $\tilde{Z}$, respectively, and $|\mathcal{G}|_{\mathcal{E}}$ denotes the total edges in $\mathcal{G}$. Therefore, if $\mathcal{G}$ exclusively contains homogeneous edges, $q(\mathcal{G}) = 0$, and if all edges are heterogeneous, then $q(\mathcal{G}) = 1$. In this work, we use the degree of edge heterogeneity, $q(\mathcal{G})$, as a measure of latent space integrity between un-shifted and shifted samples under perturbations $\mathcal{T}$. A visual interpretation of $q(\mathcal{G})$ is provided in Figure 2.

$$q(\mathcal{G}) = 1 - \frac{\left(\left|\mathcal{G}^Z\right|_{\mathcal{E}} + \left|\mathcal{G}^{\tilde{Z}}\right|_{\mathcal{E}}\right)}{|\mathcal{G}|_{\mathcal{E}}} \tag{1}$$

Figure 2: Visualization of changes in latent space between unmodified ($z$; orange) and broadband noise shifted data ($\tilde{z}$; blue) as measured by the proportion of heterogeneous edges (grey) in the Delaunay neighborhood graph (Section 3.3). Note that as noise strength increases from left to right, there is a decrease in the proportion of heterogeneous edges. This decrease in proportion is interpreted as decreasing latent space integrity and corresponds to lower $q(\mathcal{G})$ scores. Figure best viewed in color.

### 3.4 Quantifying model uncertainty using Monte Carlo dropout

In this work, we use a Monte Carlo dropout (MCD) strategy to examine per-sample variability in predictions to measure model 'uncertainty' for a new sample (24). Specifically, we directly compare the changes in model uncertainty (measured using the standard deviation) under the aforementioned EEG-DS. Here we briefly summarize the theoretical background and our implementation.

In the Bayesian setting, the predictive distribution of a model $f$ with parameters $\theta$ for an input $x$ and output $\hat{y}$ is expressed as $p(\hat{y} \mid x, D) = \int p(\hat{y} \mid x, \theta) p(\theta \mid D) d\theta$, where $D$ is training data. Because exact computation of this expression is intractable, we perform approximate inference using a variational distribution. A previous study showed that training neural networks with Bernoulli dropout variables is equivalent to approximate inference of the posterior of model parameters (24). With $T$ sets of Bernoulli realizations of $\theta$ $\{\theta_t\}_{t=1}^T$, the Monte Carlo estimate of the mean prediction and its variance are computed as $\mathbb{E}[\hat{y}|x] \approx \frac{1}{T} \sum_{t=1}^T f(x, \theta_t)$ and $Var(\hat{y}|x) \approx \frac{1}{T} \sum_{t=1}^T (f(x, \theta_t) - \mathbb{E}[\hat{y}|x])^2$. In our experiments, we used a dropout probability of $c = 0.5$ and repeated forward passes $T = 20$ times to estimate the mean and variance of predictions $\hat{y}$. We then compare the standard deviation and the agreement index of the predictions to make conclusions about the predictive uncertainty of

regression and classification tasks, respectively. The agreement index between multiple Monte Carlo dropouts for a threshold $\tau_t$ is defined as, $\phi = \frac{1}{T} \sum_{t=1}^{T} \mathbb{1}\{f(x, \theta_t) \geq \tau_t\}$.

## 4  Experimental Setup

Our overall experiment setup is illustrated in Figure 3. Using raw EEG signals and their shifted versions under transforms EEG-DS introduced in Section 3.2, we evaluate the robustness of EEG encoders in both in-sample and out-of-sample data. Through our experiments, we hope to answer the following questions: a) do latent space analysis and predictive uncertainty help anticipate model behavior in deployment settings?, b) what types of real-world EEG data shifts impact model robustness the most?, and c) do some encoders exhibit better robustness than others? In the following, we briefly describe the components of our evaluation setup: the EEG datasets, preprocessing steps, EEG feature encoders, learning tasks, data splits, and evaluation metrics.

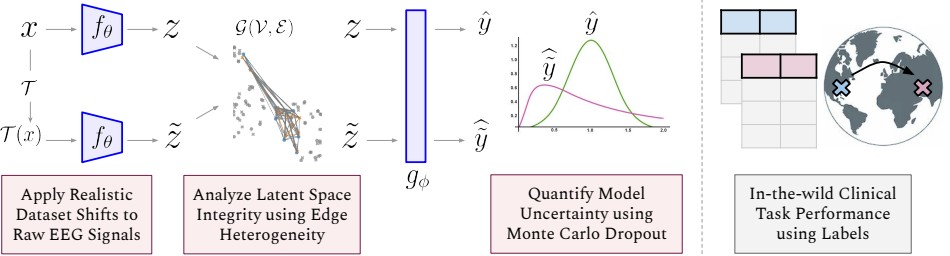

Figure 3: **The overall experiment setup.** First, we trained several EEG feature encoders ($f_\theta$) using unmodified training data in fully-supervised, self-supervised, and traditional PSD feature-based settings. Second, we generated latent representations for the unmodified test data and the modified test data (using data shifts $\mathcal{T}$), and compared the topological similarities in the latent space ($z$ and $\tilde{z}$). Next, we fine-tuned the feature encoders on classification and regression tasks ($g_\phi$) and generated predictions $y$ and $\tilde{y}$ for both $z$ and $\tilde{z}$. We then quantified the uncertainty in predictions using an MCD strategy. Finally, we evaluated encoder performances in the in-sample test data (for modified and unmodified versions) and in the out-of-sample data.

**EEG dataset & preprocessing:** We utilize EEG data from two large-scale databases, the TUH-EEG database from Temple University Hospital (13) and the NMT database from Pak-Emirates Military Hospital (26). For our experiments, we utilize 2,993 EEGs from TUH-EEG (a subset known as TUAB) and 1,369 EEGs from the NMT database, both of which contain normal and abnormal labels. All EEGs were recorded according to the standard 10-20 montage (41). We extracted age labels from text reports accompanying the TUAB recordings using a previously published tool (42). Next, we preprocessed the EEGs as follows: a) we ordered the EEG channels as 'Fp1', 'Fp2', 'F3', 'F4', 'C3', 'C4', 'P3', 'P4', 'O1', 'O2', 'F7', 'F8', 'T7', 'T8', 'P7', 'P8', 'Fz', 'Cz', and 'Pz', b) resampled the recordings to 128Hz, c) applied a bandpass filter between 0.5Hz and 45.0Hz, d) divided the recordings into contiguous non-overlapping epochs of 10-seconds each, e) identified and removed bad epochs when the total power in their 'Cz' channel exceeded 2 standard deviations as calculated from statistics of each recording, f) clipped amplitude values to not exceed $\pm 800\mu$V, and finally g) we normalized the recording per-channel using the identified valid epochs. Note that whenever transforms $\mathcal{T}$ were utilized, they were applied directly on the raw data before preprocessing.

**EEG encoders:** We train three diverse types of raw EEG encoders: fully-supervised, self-supervised, and traditional feature-based. The fully supervised encoders (FSE) were based on the widely used 'ShallowNet' architecture (15) with additional classification and regression heads described below. The self-supervised encoder (SSE) also utilized the 'ShallowNet' architecture with pre-training performed using a proxy measure of the patient's behavioral state (18). Finally, we utilized a power-spectral-density-based encoder (PSDE) with EEG spectral power calculated within 7 frequency bands: $\delta$ (2-4Hz), $\theta$ (4-8Hz), low $\alpha$ (8-10Hz), high $\alpha$ (10-13Hz), low $\beta$ (13-16Hz), high $\beta$ (16-25Hz), and $\gamma$ (25-40Hz) for each of the 19 channels. Both FSE and SSE took raw EEG traces as inputs and produced 128-dimensional embeddings, while PSDE generated 19×7=133 features.

We utilized classification and regression heads to develop task-specific predictors using the above encoders. Note that we trained FSE entirely using the task labels, whereas we fine-tuned SSE by training the classification/regression layer while keeping the encoder network frozen. In addition, we employed dropout with $p = 0.5$ in all heads and used a binary cross-entropy loss and a smooth variant

of the mean absolute error loss for training the classification and regression heads, respectively. We trained the models up to a maximum of 500 epochs and monitored convergence with 373 validation EEGs. We used either Adam or SGD optimizers with a cyclic learning rate scheduler (43). The full set of hyper-parameters is provided in the supplement and documented in the publicly released code.

**Classification and regression tasks:** We utilized meaningful classification and regression tasks to evaluate the robustness of the above encoders. EEG grade classification as 'normal' or 'abnormal' is a common task performed by clinical experts via visual review. Performing such classification using ML approaches in an automated fashion is of great interest since it can potentially reduce the workload of clinical experts. Next, the potential to predict an individual's age using EEGs, a regression task commonly known as brain age prediction, is useful because it can indicate deviations from healthy brain aging and therefore help with early diagnosis of neurodegenerative diseases. Hence, we utilized two tasks: a binary classification task classifying EEG grades as 'normal' or 'abnormal' and a regression task predicting brain age from EEGs to conduct our experiments.

**Training, validation, and testing:** We split TUAB into fixed training (1,490 EEGs; 258,631 epochs), validation (373 EEGs; 68,493 epochs), and held-out test (466 EEGs; 82,331 epochs) sets. Furthermore, we performed all model training on TUAB at the epoch level without any shifts. Since both EEG grade and age are recording-level measures, during evaluation, we aggregate epoch-level predictions by averaging the predictions across all epochs in a recording (18; 21).

**In-sample and out-of-sample testing:** We evaluated the robustness of EEG encoders in the in-sample setting by comparing performances between the modified and unmodified TUAB test set and out-of-sample (i.e., in-the-wild) performance using the external NMT dataset. We restricted NMT evaluations to data from adults (18+) to avoid any physiological variability unseen during training.

**Evaluation metrics:** We performed all evaluations on the held-out TUAB test set and the external NMT dataset. The key metrics utilized to perform the proposed encoder assessments are described in Section 3. Based on the graph construction outlined in Section 3.3, we interpreted the measure of edge heterogeneity obtained using Equation 1 as a score of the integrity of the latent space under perturbation. For each encoder, we compared the representations of raw EEG samples against their shifted versions in the latent space. Next, we used predictive uncertainty to analyze the per-sample variability of encoders under dataset shifts. Specifically, we analyzed the median agreement index ($\phi$) and standard deviation in model predictions obtained via MCD as measures of uncertainty. Finally, we quantified task performance using the ground truth labels. To analyze task performance, we used the area under the receiver operating characteristic curve (AUC) and the mean absolute error (MAE) metrics for EEG grade and brain age tasks, respectively. Note that we performed the analyses of the latent space using embeddings of all the epochs in the test set and analyzed task performance and uncertainty using aggregated EEG-recording level metrics.

**Software & hardware:** The preprocessing, baseline features, and model training were implemented using a combination of the following Python libraries: 1) MNE-python (44), 3) PyTorch (45), and 3) NumPy (46). Delaunay graph construction and analysis were performed using code released by (25). The experiments were performed using 2 Nvidia RTX 3090 GPUs.

# 5   Results

Our results are summarized in Figures 4 and 5, and Table 2. We performed latent space analysis and uncertainty quantification without using the ground truth labels and evaluated whether the changes in those measures resulting from test-time data shifts indicated performance degradation in the in-sample and out-of-sample test sets. In the following, we use the following acronyms to refer to the data shifts and encoders: BP - bandpass filter, QP - quantization precision, IN - impedance noise, BN - broadband noise, SSE - self-supervised encoder, FSE-grade - fully-supervised encoder fine-tuned on grade classification, FSE-age - fully-supervised encoder fine-tuned on age prediction, PSDE - traditional power-spectral density-based encoder.

**Impact of data-shifts on predictions**: Figure 4 shows an example of how model predictions are affected when the proposed transformations are applied to the input data. In the top row, we compare the predicted probability of EEG grade by the FSE encoder on held-out data under additive broadband noise $t_{\text{BN}}$ of varying strength $\sigma$. The probabilities are grouped along the x-axis by expert labels. We observe that the predicted probability distributions for the two ground truth classes overlap more

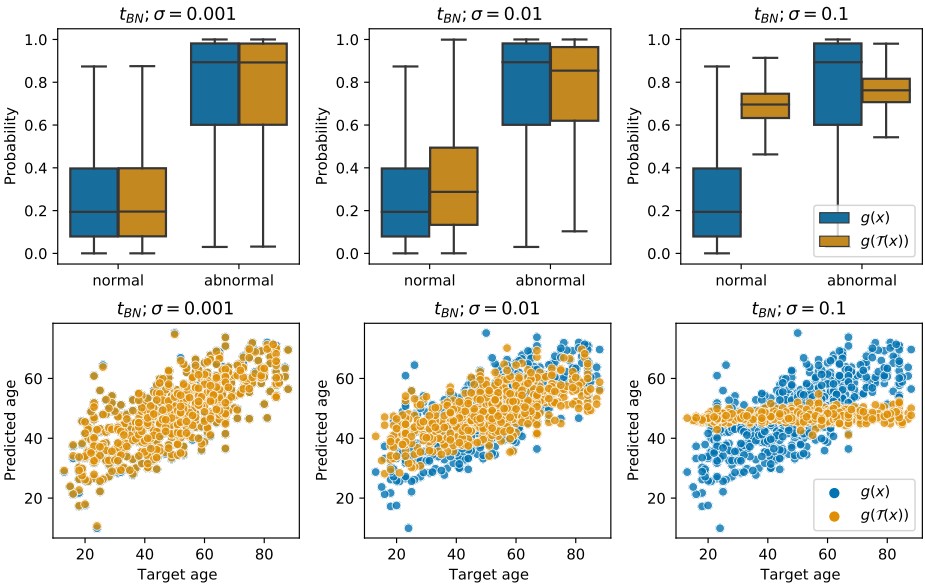

Figure 4: The impact of broadband noise on the predictions of FSE encoder in EEG grade and brain age tasks.

under stronger shifts. In the bottom row, we show scatter plots of brain age predicted by FSE against the ground truth under similar comparison settings as above. With stronger shifts, we observe a collapse in the range of FSE predictions.

**Evaluation of the latent space**: Figure 5 shows the latent space integrity scores ($\uparrow$) in the in-sample (lines) and out-of-sample ('$\times$'s) settings for multiple EEG encoders. The same results are provided in a tabular form in the supplement including standard deviations. First, we established an empirical baseline for ideal integrity scores by comparing un-modified data with itself for each encoder (labeled as 'No shift' in Figure 5). Then we compared the in-sample integrity scores between the encoders by increasing the strength of each data shift (left $\rightarrow$ right). We find that all proposed data shifts affect the integrity scores, albeit to varying degrees, with BN and QP being the most and least effective, respectively. BN with $\sigma$=0.1 is aggressive enough to yield only homogeneous edges, and the measure goes to 0. In the 'No shift (baseline)' scenario, the number of homogeneous and heterogeneous edges is roughly equal, leading to an integrity score of approximately 0.5. Interestingly, we also find that PSDE and SSE encoders produce better overall integrity in the latent space compared to others. We then computed these scores between unmodified in-sample and out-of-sample test sets for 100 randomly chosen EEGs. We find that there are notable differences between the two test sets, and that the representations generated by SSE preserve integrity the most.

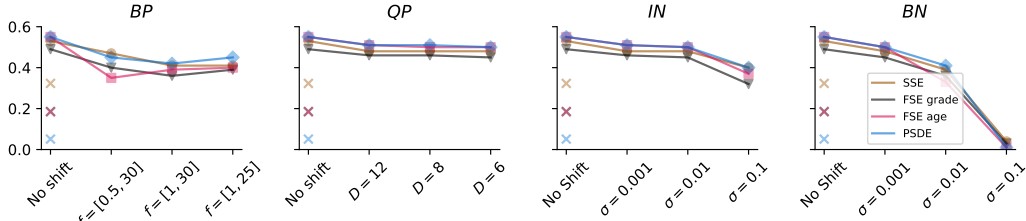

Figure 5: Integrity measures ($q(\mathcal{G})$) in the in-sample (lines) and out-of-sample ('$\times$'s) settings for different encoders. Measures were computed at different strengths of shifts in the in-sample setting, and between unmodified TUH and NMT tests for the out-of-sample setting.

**Task performance & uncertainty:** Table 2 lists the task performance measures and the uncertainty measures for EEG grade classification and brain age regression tasks evaluated on the in-sample and out-of-sample test sets under different data shifts. For additional context, the same metrics computed on the train and validation sets are provided as supplement. The task performance metrics reported are the means of the recording-level AUCs and MAEs computed based on the 20 MCD repeats (standard deviations are provided in the supplement). Regardless of shift, we find that FSE outperforms both SSE and PSDE in both tasks significantly. However, SSE and PSDE exhibit consistent performance

under perturbations in the age prediction task although their absolute performance is worse than FSE. Data shifts IN and BN affect all encoders in EEG grade classification, and BP, IN, and BN affect FSE encoder in age prediction. As measures of uncertainty, we report the average agreement index in EEG grade prediction and the standard deviation of the predicted age values across 20 MCD repeats. We find that both SSE and PSDE exhibit overconfidence in model predictions despite achieving worse task performance than FSE in both tasks. Next, all EEG-DS cause only marginal changes on uncertainty measures and only the strongest BN ($\sigma = 0.1$) causes a noticeable change. The out-of-sample evaluation shows a universal degradation in both task performance and uncertainty measures across all encoders compared to the baseline.

Table 2: Task performance and uncertainty measures for classification (EEG Grade) and regression (Age) tasks in in-sample and out-of-sample data under different shifts. Abbreviations: AUC - area under curve ($\uparrow$), MAE - mean absolute error ($\downarrow$), $\phi$ - agreement index ($\uparrow$), and SD - standard deviation in predicted values ($\downarrow$). Up/down arrows indicate the direction of desirable trends. Bold highlights indicate the scenarios when the proposed EEG shifts significantly degraded the performance.

| EEG Shifts ($\mathcal{T}$) | EEG Grade (AUC) | | | Age (MAE) | | | EEG Grade ($\phi$) | | | Age (SD) | | |
|---|---|---|---|---|---|---|---|---|---|---|---|---|
| | SSE | FSE | PSDE | SSE | FSE | PSDE | SSE | FSE | PSDE | SSE | FSE | PSDE |
| No shift (baseline) | 0.77 | 0.92 | 0.77 | 13.74 | 9.47 | 15.56 | 0.99 | 0.99 | 0.96 | 0.10 | 0.50 | 0.32 |
| No shift - NMT | 0.72 | 0.72 | 0.64 | 17.46 | 12.41 | 17.74 | 0.97 | 0.97 | 0.85 | 0.14 | 0.73 | 0.52 |
| $t_{BP}$ ($f = [0.5, 30]$) | 0.77 | 0.92 | 0.77 | 13.74 | 11.24 | 15.52 | 0.99 | 1.00 | 0.96 | 0.10 | 0.47 | 0.32 |
| $t_{BP}$ ($f = [1, 30]$) | 0.78 | 0.92 | 0.78 | 13.70 | 11.14 | 15.44 | 0.99 | 1.00 | 0.95 | 0.10 | 0.46 | 0.33 |
| $t_{BP}$ ($f = [1, 25]$) | 0.78 | 0.92 | 0.77 | 13.70 | 11.10 | 15.47 | 0.99 | 0.99 | 0.95 | 0.10 | 0.46 | 0.33 |
| $t_{QP}$ ($D = 12$) | 0.77 | 0.92 | 0.77 | 13.74 | 9.47 | 15.56 | 0.99 | 0.99 | 0.96 | 0.10 | 0.50 | 0.32 |
| $t_{QP}$ ($D = 8$) | 0.77 | 0.92 | 0.77 | 13.74 | 9.47 | 15.56 | 0.99 | 0.99 | 0.96 | 0.10 | 0.50 | 0.32 |
| $t_{QP}$ ($D = 6$) | 0.77 | 0.92 | 0.77 | 13.75 | 9.51 | 15.56 | 0.99 | 0.99 | 0.96 | 0.10 | 0.49 | 0.32 |
| $t_{IN}$ ($\sigma = 0.001$) | 0.77 | 0.92 | 0.77 | 13.74 | 9.47 | 15.56 | 0.99 | 0.99 | 0.96 | 0.10 | 0.50 | 0.32 |
| $t_{IN}$ ($\sigma = 0.01$) | 0.77 | 0.92 | 0.76 | 13.75 | 9.49 | 15.58 | 0.99 | 0.99 | 0.96 | 0.10 | 0.49 | 0.32 |
| $t_{IN}$ ($\sigma = 0.1$) | **0.71** | **0.91** | **0.67** | **13.77** | **11.17** | **16.07** | **0.99** | **0.99** | **0.98** | **0.10** | **0.44** | **0.27** |
| $t_{BN}$ ($\sigma = 0.001$) | 0.77 | 0.92 | 0.77 | 13.74 | 9.50 | 15.56 | 0.99 | 0.99 | 0.96 | 0.10 | 0.49 | 0.32 |
| $t_{BN}$ ($\sigma = 0.01$) | 0.78 | 0.91 | 0.78 | 13.84 | 11.06 | 15.41 | 0.99 | 0.99 | 0.95 | 0.08 | 0.47 | 0.31 |
| $t_{BN}$ ($\sigma = 0.1$) | **0.72** | **0.86** | **0.71** | **14.08** | **14.05** | **14.66** | **0.83** | **0.94** | **0.83** | **0.11** | **0.76** | **0.35** |

# 6  Discussion

The aim of this study was to develop model diagnostic measures for EEG feature encoders that help anticipate deployment failures during model development itself. We modeled real-world variability in EEG signals by designing four data shifts, i.e., EEG-DS (Table 1), that capture the effects of band-pass filter settings, analog-to-digital quantization precision, and manifestations of noise. Then, we trained diverse types of EEG encoders (self-supervised, fully-supervised, and conventional) on un-shifted data for two clinically important tasks (Section 4), and assessed their robustness under these shifts during model development. Notably, we analyzed the encoders' latent space (Figure 5) and predictive uncertainty (Table 2) in a label-agnostic fashion, in addition to analyzing task performance (Table 2). Finally, an evaluation using an external dataset indicated that the performance drop observed in the out-of-sample setting is proportional to what we observed in the above experiments. The significance of our study is that we propose evaluation metrics beyond traditional task performance that can be computed during model development itself to evaluate model robustness and anticipate future failures in practical deployment settings. Furthermore, we emphasize that the benefit of our approach is that it does not require access to additional data or labels from the deployment setting.

**The value of latent space analysis**: If we consider task performance as a gold-standard metric in deployment settings, the latent space analysis allows us to see 'early' signs that predict degraded task performance (Figure 5 and Table 2). The results of aggressive shifts (e.g., $\sigma$=0.1) in the in-sample experiments strongly correlated with the out-of-sample performance, highlighting the potential use of the latent space analysis. Intuitively, a strong distortion in the latent space will likely push samples from one side of the decision boundary to the other, leading to worse performance.

**The value of uncertainty quantification**: Our uncertainty studies identify a major pitfall that can occur during deployment. We found that both SSE and PSDE encoders were highly confident in their predictions, although their task performances were much worse compared to FSE. This might indicate that the SSE and PSDE encoders were not sufficiently trained to achieve desirable task performances (i.e., classifier bias) and hence showed less variability in predictions. On the other

hand, the uncertainty measures of the FSE encoder provided meaningful interpretations, although the changes in uncertainty under the proposed data shifts were marginal. We believe that analyzing measures of uncertainty in conjunction with task performance can help assess real-world robustness.

**Data shifts**: Trends in Figure 5 indicate that a (1-25Hz) band-pass filter and strong additive broadband or narrowband noise ($\sigma = 0.1$) impact latent space the most. Table 2 shows the same trend in task performance, although the effects are more pronounced in age prediction compared to EEG grade classification. From an EEG perspective, sensitivity to band-pass filtering and noise is expected since they affect the spectral content of the signal, which is important for the tasks of EEG grade classification and brain age regression. We found that the data shifts did not impact uncertainty measures all that much except for the strongest broadband noise ($\sigma = 0.1$). While we do not consider all sources of EEG shift, the fact that the proposed instrumentation-related shifts can anticipate out-of-sample performance suggests that they represent a good fraction of real-world diversity and are indeed valuable in anticipating deployment failures.

**EEG feature encoders**: A general trend seen in Table 2 across all shifts is that FSE provides better absolute performance and that SSE and PSDE encoder performances are more stable. This apparent trade-off between absolute performance and robustness can be explained by considering the effective model sizes (i.e., the number of learnable parameters) in each setting. Recall from Section 4 that only the last linear layer $g_\phi(.)$ is trainable for SSE and PSDE while the entire network $g_\phi(f_\theta(.))$ is trained in case of FSE. Due to a disproportionately higher number of parameters, FSE can be expected to perform better than SSE and FSE. However, this also implies that dropout in FSE would inactivate a disproportionately higher number of units, leading to less stable performance.

**Limitations and future directions:** Besides instrumentation-related variability, data shifts originating from physiological changes in brain activity (e.g., sleep) may also play a significant role in real-world robustness. In addition, our design of the quantizer precision shift could be made more rigorous by re-mapping the discrete signal amplitude to a restricted set of quantized numerical levels instead of manipulating decimal precision. In future work, we will implement additional data shifts representing the physiological variability in brain activity and investigate whether adversarial training and training with data shifts could improve real-world robustness.

**Data and code availability:** The TUAB and NMT datasets used in this study are already publicly available. Model checkpoints and code needed to reproduce results and scripts to apply the proposed robustness evaluation approach on other datasets are provided at `https://github.com/neerajwagh/evaluating-eeg-representations`.

## 7 Conclusion

In this paper, we developed an approach to assess the real-world robustness of EEG-ML models during model development itself. Our approach included the implementation of realistic data shifts and the evaluation of model robustness using a latent space topological analysis and quantification of predictive uncertainty. We developed four data shifts for EEG signals reflecting potential instrumentation-related variability observable in the real world. We then performed experiments to evaluate differences in latent space integrity, predictive uncertainty, and task performance under the above data shifts and correlated the results with out-of-sample (i.e., in-the-wild) performance. Within our experimental setting, including three EEG feature encoders, two downstream tasks, and two large-scale EEG datasets, our results demonstrate that the proposed approach can help anticipate real-world performance during model development. In future work, we will consider other plausible data shifts, adversarial training, and training models explicitly on realistic data shifts.

## Acknowledgments and Disclosure of Funding

This research was partially supported by the National Science Foundation (Award No. IIS-2105233), Mayo Clinic Neurology Artificial Intelligence Program, and a Mayo Clinic & Illinois Alliance Fellowship for Technology-Based Healthcare Research. The authors thank Petra Poklukar for assisting with latent space analysis.

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
