# Supplement: Evaluating Latent Space Robustness and Uncertainty of EEG-ML Models under Realistic Distribution Shifts

**Neeraj Wagh**[1*]**, Jionghao Wei**[1]**, Samarth Rawal**[1]**, Brent Berry**[2]**, Yogatheesan Varatharajah**[1,2*]
[1]University of Illinois at Urbana-Champaign   [2]Mayo Clinic

## 1  Training-related hyperparameters

Table 1: Hyperparameters related to neural network model training

| Category | Hyperparameter | Value |
|---|---|---|
| Smooth L1 regression loss | Beta threshold | 1.0 |
| Adam optimizer & Multi-step LR scheduler | Learning rate | 0.001 |
| | Betas | (0.9, 0.999) |
| | Weight decay | 1e-5 |
| | Milestones | [30, 80, 150, 200, 250, 300, 350, 400, 450] |
| | Decay factor | 0.5 |
| SGD optimizer & Cyclic LR scheduler | Base learning rate | 1e-5 |
| | Maximum learning rate | 0.01 |
| | Momentum | 0.9 |
| | Weight decay | 1e-5 |
| | Step size up | 2000 |
| | Gamma | 0.5 |
| Layer initialization (linear, conv1d, convtranspose1d) | Weight - Xavier uniform gain | 1.0 |
| | Bias - constant fill | 0.01 |

*Corresponding authors:  {nwagh2, varatha2}@illinois.edu

## 2 Variability in task performance across recordings

Table 2: Task performance for EEG grade classification (EEG Grade) and brain age regression (Age) evaluated on held-out in-sample (TUAB) and out-of-sample data (NMT) under different data shifts.

| EEG Shifts ($\mathcal{T}$) | Grade (AUC) | | | Age (MAE) | | |
|---|---|---|---|---|---|---|
| | SSE | FSE | PSDE | SSE | FSE | PSDE |
| No shift (baseline) | 0.769±0.001 | 0.920±0.001 | 0.767±0.003 | 13.742±0.004 | 9.470±0.022 | 15.560±0.015 |
| No shift - NMT | 0.717±0.002 | 0.715±0.002 | 0.638±0.009 | 17.460±0.005 | 12.411±0.017 | 17.735±0.014 |
| $t_{\mathrm{BP}}$ ($f = [0.5, 30]$) | 0.771±0.001 | 0.920±0.001 | 0.769±0.002 | 13.735±0.004 | 11.239±0.021 | 15.520±0.019 |
| $t_{\mathrm{BP}}$ ($f = [1, 30]$) | 0.778±0.001 | 0.922±0.001 | 0.776±0.004 | 13.704±0.005 | 11.144±0.018 | 15.440±0.015 |
| $t_{\mathrm{BP}}$ ($f = [1, 25]$) | 0.779±0.001 | 0.921±0.001 | 0.774±0.005 | 13.697±0.005 | 11.100±0.018 | 15.468±0.016 |
| $t_{\mathrm{QP}}$ ($D = 12$) | 0.769±0.001 | 0.920±0.001 | 0.767±0.003 | 13.742±0.004 | 9.470±0.022 | 15.560±0.015 |
| $t_{\mathrm{QP}}$ ($D = 8$) | 0.769±0.001 | 0.920±0.001 | 0.767±0.003 | 13.742±0.004 | 9.470±0.022 | 15.560±0.015 |
| $t_{\mathrm{QP}}$ ($D = 6$) | 0.769±0.001 | 0.920±0.000 | 0.767±0.003 | 13.747±0.004 | 9.507±0.021 | 15.558±0.015 |
| $t_{\mathrm{IN}}$ ($\sigma = 0.001$) | 0.769±0.001 | 0.920±0.001 | 0.767±0.003 | 13.742±0.004 | 9.469±0.022 | 15.560±0.015 |
| $t_{\mathrm{IN}}$ ($\sigma = 0.01$) | 0.767±0.001 | 0.920±0.000 | 0.762±0.003 | 13.745±0.004 | 9.490±0.022 | 15.575±0.014 |
| $t_{\mathrm{IN}}$ ($\sigma = 0.1$) | 0.708±0.001 | 0.905±0.001 | 0.669±0.002 | 13.766±0.005 | 11.169±0.019 | 16.072±0.013 |
| $t_{\mathrm{BN}}$ ($\sigma = 0.001$) | 0.770±0.001 | 0.920±0.001 | 0.768±0.003 | 13.742±0.004 | 9.504±0.022 | 15.556±0.015 |
| $t_{\mathrm{BN}}$ ($\sigma = 0.01$) | 0.784±0.001 | 0.913±0.001 | 0.779±0.003 | 13.835±0.003 | 11.058±0.021 | 15.408±0.016 |
| $t_{\mathrm{BN}}$ ($\sigma = 0.1$) | 0.718±0.011 | 0.856±0.004 | 0.709±0.012 | 14.075±0.004 | 14.048±0.036 | 14.662±0.013 |

# 3 Comparison of metrics between train, validation, and test data splits

Table 3: Task performance and uncertainty measures for EEG grade classification (EEG Grade) and brain age regression (Age) evaluated on the fixed TUAB dataset splits under different data shifts. We measure classification performance using AUC ($\uparrow$), regression accuracy using MAE ($\downarrow$), classification predictive uncertainty using the agreement index - $\phi$ ($\uparrow$), and regression predictive uncertainty using the standard deviation of the predicted values - SD ($\downarrow$). Up/down arrows indicate favorable direction.

| EEG Shifts ($\mathcal{T}$) | EEG Grade (AUC) | | | Age (MAE) | | | EEG Grade ($\phi$) | | | Age (SD) | | |
|---|---|---|---|---|---|---|---|---|---|---|---|---|
| | SSE | FSE | PSDE | SSE | FSE | PSDE | SSE | FSE | PSDE | SSE | FSE | PSDE |
| No shift - TUAB test | 0.77 | 0.92 | 0.77 | 13.74 | 9.47 | 15.56 | 0.99 | 0.99 | 0.96 | 0.10 | 0.50 | 0.32 |
| No shift - TUAB val | 0.78 | 0.91 | 0.77 | 12.79 | 8.69 | 14.40 | 0.99 | 0.99 | 0.95 | 0.10 | 0.49 | 0.32 |
| No shift - TUAB train | 0.77 | 0.96 | 0.76 | 14.31 | 5.85 | 45.74 | 0.99 | 0.99 | 0.95 | 0.10 | 0.50 | 0.31 |

Note that 'No shift - TUAB test' is the same result reported in Table 2 of the main text under 'No shift (baseline)'. With the exception of task performance metrics for the FSE and PSDE encoders on the train set, in all other cases, the metrics computed on the three dataset splits are in close alignment.

# 4 Variability in latent space integrity scores across recordings

The latent space analysis was conducted on a per-recording basis. Therefore, the variability reported in Table 4 is across multiple recordings in the held-out TUAB set.

Table 4: Variability in latent space integrity scores (↑) computed between unmodified and various realistic shifted versions of EEG data (rows) for multiple EEG encoders (columns).

| EEG Shifts ($\mathcal{T}$) | EEG Encoders ($f_\theta$) | | | |
|---|---|---|---|---|
| | SSE | FSE-Grade | FSE-Age | PSDE |
| No shift (baseline) | 0.53±0.13 | 0.49±0.18 | 0.55±0.07 | 0.55±0.06 |
| $t_{\text{BP}}$ ($f = [0.5, 30]$) | 0.47±0.11 | 0.40±0.19 | 0.35±0.15 | 0.45±0.05 |
| $t_{\text{BP}}$ ($f = [1, 30]$) | 0.41±0.13 | 0.39±0.21 | 0.40±0.14 | 0.45±0.06 |
| $t_{\text{BP}}$ ($f = [1, 25]$) | 0.41±0.14 | 0.36±0.22 | 0.39±0.14 | 0.42±0.11 |
| $t_{\text{QP}}$ ($D = 12$) | 0.48±0.11 | 0.46±0.16 | 0.50±0.05 | 0.51±0.02 |
| $t_{\text{QP}}$ ($D = 8$) | 0.48±0.11 | 0.46±0.16 | 0.50±0.05 | 0.51±0.02 |
| $t_{\text{QP}}$ ($D = 6$) | 0.48±0.11 | 0.45±0.16 | 0.50±0.06 | 0.50±0.04 |
| $t_{\text{IN}}$ ($\sigma = 0.001$) | 0.48±0.11 | 0.46±0.16 | 0.51±0.04 | 0.51±0.02 |
| $t_{\text{IN}}$ ($\sigma = 0.01$) | 0.48±0.11 | 0.45±0.17 | 0.50±0.07 | 0.50±0.03 |
| $t_{\text{IN}}$ ($\sigma = 0.1$) | 0.40±0.18 | 0.32±0.24 | 0.37±0.18 | 0.40±0.15 |
| $t_{\text{BN}}$ ($\sigma = 0.001$) | 0.48±0.12 | 0.45±0.17 | 0.50±0.06 | 0.50±0.03 |
| $t_{\text{BN}}$ ($\sigma = 0.01$) | 0.39±0.19 | 0.36±0.21 | 0.33±0.18 | 0.41±0.15 |
| $t_{\text{BN}}$ ($\sigma = 0.1$) | 0.04±0.09 | 0.03±0.07 | 0.02±0.04 | 0.01±0.03 |