# OpenReview forum: "Evaluating Latent Space Robustness and Uncertainty of EEG-ML Models under Realistic Distribution Shifts"
_NeurIPS.cc/2022/Conference — NeurIPS 2022 Accept_

### Official Review · Reviewer_VEzQ · 2022-07-10

**Rating:** 7
**Confidence:** 5
**Soundness:** 4 excellent
**Presentation:** 4 excellent
**Contribution:** 3 good

**Summary:**

The paper provides a description how four types of feasible and realistic dataset shifts can be used to estimate their effect on robustness of the classification and regression tasks in EEG analysis.

The paper uses a latent space of features extracted from the EEG signals and compares how the latent space data points of the original and the transformed data  create a Delaunay graph. If all edges of the in the Delaunay (i.e nearest neighbors)  in the data set would contain only edges from the original latent space points  to the corresponding transformed  latent space points, one would consider the perturbations to be robust.

The distributions of the outputs for the original and the transformed data are created with Monte Carlo dropout using the model trained by the original model. This way the data shift  effect to the output  distributions can be seen and quantified for robustness estimation. Experiments are performed, for example, with different sizes of added noise and it is seen that noisy data will lead to collapse of the regression task.

The authors suggest that following the described analysis of the robustness can be used as indication of problems in generalization capability of the model under realistic data shifts in medical domain.

**Questions:**

To have a more holistic view of the robustness of the trained solution with the use of a given, extensive, data set, one should also check how adversarial training would improve the generalization under the four feasible transformations. This would check and answer if the expected realistic data shifts would be covered automatically by the adversarial robustness. Now, the trained version used is not the optimal model  to estimate the robustness that is achievable by the available data set.

Also, the result estimation of the robustness reached by augmenting the data with the four known transformations should be performed as a comparison, as it is interesting to see to what extent the knowledge of the possible data shifts that is used in the training would alleviate the problem.

This paper  a very good and thorough work, but more aspects should be covered. Questions to be answered for better rating:
1) What happens for the generalization capability if adversarially robust training is used?
2) What happens for the generalization capability if domain knowledge augmentation (with the four types) is used?

I would appreciate a discussion on these points with the related additional testing.

**Limitations:**

The paper is a excellent step forward to address the data shifts in the medical domain

**Strengths And Weaknesses:**

The paper is well written and understandable. It provides insights on a long standing problem of training systems with data that originate from restricted domains that are then deployed in other domains. Of course, the ideal case would be to alleviate the problem with big enough data sets that would already contain all the domains. However, even this would not solve the problem for data shifts that happen in time. The problem is highly relevant.

The paper uses four different realistic transformations, but is not using these to augment the data in the training phase. In many cases augmenting the training data significantly improves the results. Keeping the training data set intact by principle does not bring extra value, because the final verdict really comes from a properly gathered test data that contains the variability of data from different sources and domains. These should cover the data shifts. Using a held out data for testing the generalization is not, in my opinion, a good way to test an AI model in the medical domain.

Also, using only four transformations is quite small set of variation to the signal. One could argue that it will not provide a comprehensive set of variations that the system will encounter. Using adversarial robustness in training would automatically find the problematic dimensions and improve robustness for a larger, more comprehensive set of  perturbations.

Now, the value of the paper now lies in the analysis of what is happening if the realistic transformations would not be known - and indicate the failures in robustness when this would happen. Choosing these to be feasible actually minimizes the problem, as one may expect that these changes would already occur also inside the original training data, for example in variations of the electrode impedance. Hence, these would be covered by the natural variation. I have some suggestion below to increase the impact of the paper by further analysis.

---

> ### Author Response · Authors · 2022-08-02
> **Response to Reviewer VEzQ**
>
> We thank the reviewer for the constructive feedback, which we will incorporate in the revised version.
>
> ### Comments on data diversity –
>
> We agree with the reviewer that ideal training and test sets would capture the diversity that is pervasive in the real world. Unfortunately, in the case of EEG, despite the urgent need for reliable predictive decision support, this real-world diversity is not accessible to model developers. Instead, the EEG-ML community has so far relied on a single publicly accessible repository (TUH, the dataset used in our study) that is collected from one hospital/institution. While the volume of TUH data helps in gaining robustness to neurological/physiological variability, there is an outstanding need for assessing robustness to acquisition/hardware variability.
>
> ### Train-time shifts and adversarial robustness –
>
> Thank you for these suggestions. As pointed out by the reviewer, training with the known shifts may improve generalization performance and is an important future direction for this work.
>
> We would also like to thank the reviewer for pointing us to the interesting area of adversarial learning that we had not considered earlier in this study. While adversarial training may improve robustness to a larger set of perturbations, it is unclear whether these perturbations would reflect realistic variability. However, this hypothesis would be interesting to confirm via experiments. Nevertheless, in sensitive decision-making settings such as healthcare, a reasonable first step is to ensure model robustness to near-certain known shifts. Our work aids in this type of assessment/model audit.
>
> Due to time constraints, we performed a simpler experiment to demonstrate the value of the proposed approach. We evaluated our trained encoders for the same tasks on a large external EEG dataset (NMT) which was recently made publicly available. NMT data was collected in a different country with different EEG hardware, making it a sufficiently diverse test set. We note that we restricted the evaluations to data from adults (18+) to avoid physiological variability not seen during training. The task performance results are shown under ‘No shift – NMT (in-the-wild)’ in Table 2 (https://figshare.com/s/f90b39e763e9a0b92f83?file=36519543). We have also included these results in figure 3 to illustrate how the latent space integrity changes in the original test set under the proposed transformations reflect in-the-wild scenario (https://figshare.com/s/f90b39e763e9a0b92f83?file=36519540).
>
> We observe a universal degradation in performance across all encoders. Very interestingly, aggressive test-time shifts (sigma=0.1, for example) in our previous experiments had already indicated this potential failure providing similar numbers. Although it is possible that there are additional variabilities in the new NMT dataset that we have not considered, these results suggest that the shifts that we proposed represent a good fraction of real-world diversity and are indeed valuable in anticipating deployment failures. It would be exciting to see if this performance deficit can be bridged with train-time shifts. Regrettably, while we sincerely appreciate the reviewer’s suggestion, this would require time-consuming model retraining and comprehensive evaluations that are best performed during a follow-up study.

---

### Official Review · Reviewer_PL6S · 2022-07-12

**Rating:** 3
**Confidence:** 3
**Soundness:** 2 fair
**Presentation:** 3 good
**Contribution:** 3 good

**Summary:**

This work aims to provide a holistic analysis of the robustness of EEG data representations after domain-guided data shifts. The authors provides two evaluation metrics to probe the robustness of EEG, namely latent space integrity and predictive uncertainty, on representations produced by different encoder methods.

**Questions:**

- I would like some clarifications about Figure 3 under scenario "no shift" - according to section 3.3, when all edges are homogeneous (i.e., connecting points belong to the same set), the latent space integrity measure value would be 0, but this is not as suggested in Fig 3.

**Limitations:**

Yes.

**Strengths And Weaknesses:**

Strength:
- The overall idea is well-motivated and addresses a unique and important challenge in ML for health.
- The authors performed very thorough analysis in general, testing various data shift and encoder methods - remarkable effort!

Weakness:
- The problem of evaluating representation/prediction learning accuracy under health data shift is a wide topic, and in many cases it means retaining performance across different sites/demographic groups/out-of-distribution cases. While the authors focus on four EEG device-wise data shift scenarios, the title and the introduction part should be careful not to over-claim the scope of this study.
- The highlights in Table 2 is misleading - when several numbers are equal, they should all be highlighted instead of just highlighting the numbers favoring an argument.
- The results are not strong enough for a conclusive analysis or novel insights (numbers reported in Table 2 do not show great difference).

---

> ### Author Response · Authors · 2022-08-02
> **Response to Reviewer PL6S**
>
> We thank the reviewer for the constructive feedback, which we will incorporate in the revised version.
>
> ### Title and introduction claims –
>
> Thank you for this observation. We acknowledge that certain sections in the initial submission need to be revised to reflect the study's scope better. We will revise the submission accordingly.
>
> ### Table 2 highlights misleading –
>
> We apologize for the confusion. The highlights indicate the scenarios when the proposed EEG shifts significantly degraded the performance of the ML models (i.e., sigma=0.1 for both broadband noise and impedance noise). While some numbers are similar, other metrics show significant differences. We will clarify this in the revised submission.
>
> ### Results are not strong enough –
>
> We apologize for the misunderstanding. Our results do indicate that proposed shifts are causing the ML models’ performance to degrade significantly, which are highlighted in Table 2. The measures we proposed, such as latent space integrity, reflect model performance under the proposed shifts and, therefore, can help anticipate failure modes of the ML models. Furthermore, with an additional external dataset, we now show that in-the-wild performance is similar to what was predicted by the proposed shifts (https://figshare.com/s/f90b39e763e9a0b92f83?file=36519543). We believe that these results are compelling as they demonstrate the practical value of the proposed approach.
>
> ### Latent space integrity measure in ‘No shift’ scenario -
>
> That is a good point, and we apologize for the lack of clarity. Please refer to figure 1 in the supplementary document for an illustration of how the composition of edges (and subsequently the integrity score) changes with varying degrees of dataset shift. Note that while the number of heterogeneous (gray) edges apparently increases, their proportion relative to total edges, in fact, decreases. This decrease in proportion lowers the integrity measure. Out of the shifts shown in Figure 3, broadband noise (BN; sigma=0.1) is aggressive enough to yield only homogeneous edges, and the measure does indeed go to 0. In the ‘No shift’ scenario, the number of homogeneous and heterogeneous edges is roughly equal, leading to an integrity score of approximately 0.5.

---

### Official Review · Reviewer_mwZ8 · 2022-07-12

**Rating:** 7
**Confidence:** 3
**Soundness:** 3 good
**Presentation:** 3 good
**Contribution:** 3 good

**Summary:**

The authors motivate the need for better evaluation methods for medical deep learning models, especially methods that do a better job of predicting in-the-wild problems of a model. Given the potential side-effects that an erroneous prediction can have in medicine, it is vital that better ways to predict how a model under distribution shifts of various intensities would perform, or at the very least, the cases in which the behaviour of the model would differ from the one in training and development validation/testing.

The authors firstly motivate, using domain expertise, four scalp EEG data shifts that would do a good job of capturing some real world variation of such datasets. Then they propose using latent space integrity methods and predictive uncertainty as two ways of evaluating the behaviour of a model under varying types and intensities of the four proposed dataset shifts.

The authors then proceed to apply their proposed evaluation measures on a number of large-scale pretrained medical encoders and showcase that their methods have some predictive utility over potential pitfalls of such models.

**Questions:**

My questions have already been stated in the previous section, but I will restate here for ease of access:

- Clearer correlation figures: Figures directly showing any meaningful correlations between in-the-wild test performance and the proposed methods attempting to predict issues in such performance. The authors state that lower is better in prediction uncertainty -- that seems misleading. Could you explain that statement a bit more?
- Clarity of training: Are any of the models trained *with* the proposed data shifts? What happens when they are? Does it improve performance on any of the heuristics and/or actual in-the-wild performance? I would rephrase the training section to ensure the reader knows what the models are trained on, in terms of data shifts.
- Prediction uncertainty discussion: I think it is important for the authors to talk about what prediction uncertainty could be showing here. Yes, if a model is more confident but performs worse, then that's an issue, but wouldn't a better measure be how the training/validation uncertainty changes? Ideally, to reduce any unpredictable model behaviour one would like the model to behave similarly in both data distributions. Furthermore, while the authors compare test performance across different data shifts, they do not show how the training performance of the model looks like. I believe that the training performance of the model should be compared a lot more with the testing performance, such that whatever heuristics are computed will take into account behaviour shifts from a 'known' to an 'unknown' domain in multiple degrees (i.e. from train to val/test, and from val/test to data shifted val/test and other pertubations).

**Ethics Review Area:**

["I don’t know"]

**Limitations:**

The authors have made a fair effort of addressing the limitations of their work, but I think that some of the conclusions drawn were a bit on the overclaiming end, for example:

- we developed four domain-guided data-shifts for EEG signals that reflect the real-world variability observable during test time
Yes, perhaps it covers some, but not all. That should be made more explicit.

- Through evaluation of multiple EEG feature encoders using large-scale EEG data, we empirically showed that the proposed approach can help in anticipating failure scenarios during deployment
This was done on a small set of datasets and tasks and using limited models. This should be integrated in this conclusion such that it does not mislead the reader.

**Strengths And Weaknesses:**


Originality:

The work proposed in this paper is one of high utility, and depending on ones definition of originality, could be assigned variable levels of originality. Given that this paper addresses medical deep learning, and proposes some very reasonable and relatively useful means of evaluating in-the-wild performance of models, I would consider it as original work, but not extremely so

I think this work might also enable a string of works targeting this direction which is something the medical deep learning community is definitely in need of.

(Score: 7/10)

Quality:

The quality of the work is very high, both in terms of execution and presentation.

(Score: 8/10)

Clarity:

The paper is quite clear, but could do with some additional figures showing correlations between in-the-wild performance and the proposed measures for a very direct way of showing to the reader what is going on.

(Score: 7/10)

Significance:

The ideas in this paper are of high significance, as they start a research direction towards good heuristics for in-the-wild performance for medical deep learning models. They also provide a set of what are effectively four data augmentation methods particularly suited for scalp EEG data, which could benefit the EEG community in general, and provide inspiration to others.

(Score: 7/10)

Strengths: Tackles an important problem with novel solutions and excellent execution with high degree of attention to detail. Provides interesting figures and tables to draw conclusions from.

Weaknesses:

- More datasets/tasks: It would make the claims made over the usefulness of the performance predictive methods more robust if more datasets/tasks were carried out and evaluated.
- Clearer correlation figures: Figures directly showing any meaningful correlations between in-the-wild test performance and the proposed methods attempting to predict issues in such performance.
- Clarity of training: Are any of the models trained *with* the proposed data shifts? What happens when they are? Does it improve performance on any of the heuristics and/or actual in-the-wild performance? I would rephrase the training section to ensure the reader knows what the models are trained on, in terms of data shifts.
- Prediction uncertainty discussion: I think it is important for the authors to talk about what prediction uncertainty could be showing here. Yes, if a model is more confident but performs worse, then that's an issue, but wouldn't a better measure be how the training/validation uncertainty changes? Ideally, to reduce any unpredictable model behaviour one would like the model to behave similarly in both data distributions. Furthermore, while the authors compare test performance across different data shifts, they do not show how the training performance of the model looks like. I believe that the training performance of the model should be compared a lot more with the testing performance, such that whatever heuristics are computed will take into account behaviour shifts from a 'known' to an 'unknown' domain in multiple degrees (i.e. from train to val/test, and from val/test to data shifted val/test and other pertubations).

---

> ### Author Response · Authors · 2022-08-02
> **Response to Reviewer mwZ8**
>
> We thank the reviewer for the constructive feedback, which we will incorporate in the revised version.
>
> ### Originality -
>
> We acknowledge that our work does not present a core theoretical contribution to the field. However, we note that proposed approach to assess model robustness is novel and our empirical evaluations demonstrate the practical value of the approach.
>
> ### Correlations with in-the-wild performance –
>
> Thank you for this suggestion. We performed an experiment to quantify in-the-wild performance using an external dataset (NMT). We note that the age distributions of the two datasets are different, which we did not consider as a data shift in the proposed approach as we focused primarily on instrumentation-related shifts. When corrected for the age distributions by limiting to adult populations, the in-the-wild performance in NMT was similar to the TUAB test performance with the strongest shifts (sigma=0.1, https://figshare.com/s/f90b39e763e9a0b92f83?file=36519543). This is reasonable because NMT data was collected in a different country with different EEG hardware, presenting a truly in-the-wild scenario with expected instrumentation-related differences. This is reflected in the integrity measures as shown in Figure 3 (https://figshare.com/s/f90b39e763e9a0b92f83?file=36519540).
>
> ### Additional comparisons/tasks/datasets –
>
> We evaluated an external dataset and reported results as mentioned above. Time permitting, we will add results for additional tasks.
>
> ### Clarity on training –
>
> We apologize for the lack of clarity in the training setup. The training was performed without any shifts. The shifts listed in Table 2 are only applied during test-time after all the models had been trained. We agree that training with these shifts and observing its effects on metrics is a natural extension of the current work and will strongly consider this experiment in a future study.
>
> ### Training/validation performance and uncertainty –
>
> This is a good suggestion. We note that the train, val, and test sets were all drawn from the same dataset and hence the train/val numbers were similar to the test set when evaluated without the proposed dataset shifts. Hence, we compared the metrics on the test set both with and without the proposed dataset shifts to evaluate the effect of those shifts. We will add the train/val results to the supplementary results of the revised submission.
>
> ### Overclaimed conclusions –
>
> Thank you for this observation. We will revise our conclusions.

---

> > ### Comment · Reviewer_mwZ8 · 2022-08-07
> > **Response to rebuttal**
> >
> > >Correlations with in-the-wild performance –
> > >Thank you for this suggestion. We performed an experiment to quantify in-the-wild performance using an external dataset (NMT). We note >that the age distributions of the two datasets are different, which we did not consider as a data shift in the proposed approach as we >focused primarily on instrumentation-related shifts. When corrected for the age distributions by limiting to adult populations, the in-the->wild performance in NMT was similar to the TUAB test performance with the strongest shifts (sigma=0.1, >https://figshare.com/s/f90b39e763e9a0b92f83?file=36519543). This is reasonable because NMT data was collected in a different country >with different EEG hardware, presenting a truly in-the-wild scenario with expected instrumentation-related differences. This is reflected in >the integrity measures as shown in Figure 3 (https://figshare.com/s/f90b39e763e9a0b92f83?file=36519540).
> >
> > Thank you for adding some in-the-wild performance results. Would be very useful to see more of these in the future.
> >
> > >Clarity on training –
> > >We apologize for the lack of clarity in the training setup. The training was performed without any shifts. The shifts listed in Table 2 are only >applied during test-time after all the models had been trained. We agree that training with these shifts and observing its effects on metrics >Is a natural extension of the current work and will strongly consider this experiment in a future study.
> >
> > Excellent. I look forward to such future work.
> >
> > >Training/validation performance and uncertainty –
> > >This is a good suggestion. We note that the train, val, and test sets were all drawn from the same dataset and hence the train/val numbers >were similar to the test set when evaluated without the proposed dataset shifts. Hence, we compared the metrics on the test set both with >and without the proposed dataset shifts to evaluate the effect of those shifts. We will add the train/val results to the supplementary results >of the revised submission.
> >
> > OK, yes, I understand that. I also asked if you could comment further on your thoughts about what the measure of uncertainty really means here, see below comment
> >
> > >The authors state that lower is better in prediction uncertainty -- that seems misleading. Could you explain that statement a bit more?

---

> > > ### Author Response · Authors · 2022-08-08
> > > **Response to Reviewer mwZ8**
> > >
> > > Thank you for your response.
> > >
> > > > The authors state that lower is better in prediction uncertainty -- that seems misleading
> > >
> > > We assume that the reviewer is referring to the Table 2 caption. We apologize for not explaining the interpretation of predictive uncertainty measures further.
> > >
> > > As correctly pointed out by the reviewer, the ‘higher/lower is better’ interpretation may be misleading or inconclusive if considered in isolation. For example, low standard deviation of Monte Carlo dropout predictions (SD in Table 2) could reflect both: 1) catastrophic model collapse, i.e., a constant output regardless of input, as well as 2) precise predictions for every input. Indeed, measures of predictive uncertainty must be assessed in conjunction with task performance when choosing between multiple potentially deployable models. We will clarify this in the revised submission.
> > >
> > > We hope this clarifies the reviewer’s point.

---

### Official Review · Reviewer_x119 · 2022-07-17

**Rating:** 4
**Confidence:** 3
**Soundness:** 2 fair
**Presentation:** 3 good
**Contribution:** 1 poor

**Summary:**

This paper introduces four EEG signal transformations to model the real-world variability observable during deployment. Then, the paper proposes a multi-pronged approach to evaluate the robustness of healthcare ML models. It is well-organized and easy to follow. The extensive experiments demonstrate the paper's claims.

**Questions:**

1. the paper's title is so broad. it should be focused on a EEGs.
2. the paper should list baseline models' performance on the related tasks.

**Ethics Review Area:**

["Privacy and Security (e.g., consent)"]

**Limitations:**

1. the novelty of the paper is weak, as it is adapted from an existing model and a Monte Carlo dropout-based method.
2. The title is to talk about health data representations, but the abstract and introduction talk about specific data - EEGs.
3. There should be baseline models in the experiments to be compared to demonstrate the proposed model efficiency.

**Strengths And Weaknesses:**

1. This paper introduces four EEG signal transformations to model the real-world variability observable during deployment.
2. The paper proposes a multi-pronged approach to evaluate the robustness of healthcare ML models.
3. Realistic data shifts in EEG data are explored in the paper.
4. The extensive experiments are conducted on real-world datasets and demonstrate the paper's claims.

---

> ### Author Response · Authors · 2022-08-02
> **Response to Reviewer x119**
>
> We thank the reviewer for the constructive feedback, which we will incorporate in the revised version.
>
> ### Title is too broad -
>
> We acknowledge that the title of the paper needs to be revised to reflect more accurately our focus on the neuro-EEG domain, instrumentation-related EEG shifts, and the experimental evidence presented. Nonetheless, we note that our approach to anticipate in-the-wild performance, including the design of real-world data-shifts, latent space analysis, and predictive uncertainty, is broadly applicable to other healthcare ML applications.
>
> ### Novelty –
>
> We regret that the novelty of the work was not clearly communicated in the initial submission. The problem setting our study considers, namely the assessment of model robustness during model development without access to external data, is in itself a novel idea. In addition, our study makes several original contributions: a) the design of domain-guided EEG data shifts, b) a multi-pronged approach to assess model robustness, and c) empirical results using multiple EEG ML models. The aim of this paper was not to make a theoretical contribution, rather we proposed a practical approach to tackle an issue that many healthcare ML models face currently.
>
> ### Baseline models –
>
> We apologize for the confusion. The aim of our study was to evaluate multiple EEG feature encoders under various degrees of the proposed test-time dataset shifts. Therefore, we consider baseline as the scenario where all encoders are evaluated with no dataset shift, which we already reported in Table 2.

---

### Author Response · Authors · 2022-08-08
**Revised submission**

We thank all reviewers for their feedback on the initial submission.

We have incorporated their suggestions into the revised manuscript, and the changes are highlighted in blue.

---

### Meta-Review · Area_Chair_k4XJ · 2022-08-26

**Recommendation:** Accept
**Confidence:** Certain

**Metareview:**

The paper introduces model-agnostic ways of quantifying predictive uncertainty and latent space differences in the situation when the distribution of data encountered during deployment differs from what the system was trained on and there is no access to the data itself. The model is evaluated on large scale EEG data.

The paper solves an important problem, in particular to the field of healthcare which has previously seen instances of models underperforming significantly at the time of deployment. As mentioned by the reviewers, this direction has not been sufficiently explored, so there is novelty in the problem itself, as well as in the solution. The experiments on EEG data were seen as convincing by the reviewers, though some questions were raised about the scope of the paper.

Overall, there is considerable merit in the work and recommend acceptance of this paper. In the camera ready, the authors should make sure not to overstate the applicability of their method. While this work could, in theory, be applied (or adapted) to other data, the merits of it outside of models trained on EEG data have not been demonstrated and should therefore not be stated as a given.

Reviewers x119 and PL6S have not engaged in the discussion although the authors responded to the issues they raised, which I kept in mind when issuing my recommendation.

**Award:**

No

---

### Decision · Program_Chairs · 2022-09-14

Accept